# Emissions of methane from coal thermal power plants and wetlands and its implications on atmospheric methane across the South Asian region

Mahalakshmi Venkata Dangeti[1], Mahesh Pathakoti*[1], Kanchana Lakshmi Asuri[1], Sujatha Peethani[2], Ibrahim Shaik[1], Rajan Sundara Krishnan[3], Vijay Kumar Sagar[4], Raja Pushapanathan[5], Yogesh Kumar Tiwari[4], and Prakash Chauhan[1]

[1]National Remote Sensing Centre (NRSC), Indian Space Research Organisation (ISRO), Hyderabad, India-500037
[2]Formerly at The International Center for Agricultural Research in the Dry Areas
[3]Lab for Spatial Informatics, International Institute of Information Technology (IIIT), Hyderabad-5000084, India
[4]Indian Institute of Tropical Meteorology (IITM), Pune, India-411008
[5]ICAR-Indian Institute of Soil and Water Conservation, Research Centre, Koraput, Odisha, India-763002
*Corresponding author: mahi952@gmail.com

**Abstract.** Atmospheric methane ($CH_4$) is a potent climate change agent responsible for a fraction of global warming. The present study investigated the spatio-temporal variability of atmospheric column-averaged ($X$) $CH_4$ concentrations using Greenhouse gases Observing SATellite (GOSAT) and TROPOspheric Monitoring Instrument onboard the Sentinel -5 Precursor (S5P/TROPOMI) data from 2009 to 2022 over the South Asia region. During the study period, the long-term trends in $XCH_4$ increased from 1700 ppb to 1950 ppb with an annual growth rate of 8.76 ppb year$^{-1}$. Among all natural and anthropogenic sources of $CH_4$, the rate of increase in $XCH_4$ was higher over the coal site at about 10.15$\pm$0.55 ppb year$^{-1}$ (Paschim Bardhaman) followed by Mundra ultra mega power plant at about 9.72$\pm$0.41 ppb year$^{-1}$. Most of the wetlands exhibit an annual trend of $XCH_4$ more than 9.50 ppb year$^{-1}$, with a minimum rate of 8.72$\pm$0.3 ppb year$^{-1}$ over Wular Lake. The WetCHARTs-based emissions of $CH_4$ from the wetlands were minimal during the winter and pre-monsoon seasons. Maximum $CH_4$ emissions were reported during the monsoon with a maximum value of 23.62$\pm$3.66 mg m$^{-2}$ month$^{-1}$ over the Sundarbans wetland. For the 15 Indian Agroclimatic zones, significant high emissions of $CH_4$ were observed over the Middle Gangetic Plains, Trans Gangetic Plains, Upper Gangetic Plains, East Coast Plains & Hills, Lower Gangetic Plains, and East Gangetic Plains. Further, the bottom-up anthropogenic $CH_4$ emissions data are mapped against the $XCH_4$ concentrations, and a high correlation was found in the Indo-Gangetic Plains region, indicating the hotspots of anthropogenic $CH_4$.

**Keywords:** GOSAT, S5P/TROPOMI, Column-averaged $CH_4$, South Asia, spatio-temporal, anthropogenic emissions.

## 1 Introduction

Atmospheric methane ($CH_4$) is one of the high-potential greenhouse gases (GHG) and plays a vital role in the chemistry of the atmosphere. In the troposphere, $CH_4$ oxidation is due to hydroxyl (OH) radical and produces carbon monoxide, carbon dioxide, and ozone in the presence of increased amounts of oxides of nitrogen. In contrast, in the stratosphere, oxidation of

$CH_4$ is by OH radical, atomic oxygen and chlorine (Nair and Kavitha, 2020). $CH_4$ has enormous potential for global warming, about 28 times that of $CO_2$ over 100 years (IPCC, 2021), and a comparatively short perturbation lifespan of about 12 years (Balcombe et al., 2018). Over the past decade, the research community has become more interested in anthropogenic $CH_4$ concentration due to its persistent rise in the atmosphere, and lack of knowledge regarding its source or sink (Huang et al., 2015). The long-term $CH_4$ observations from the National Oceanic and Atmospheric Administration (NOAA) have shown

a yearly increase of 8 ppb year$^{-1}$, while Shadnagar, an Indian site, shows an increase of 10 ppb year$^{-1}$ (Sreenivas et al., 2022). Though the emissions have increased over the past 20 years, the causes remain unclear. Recent research suggests that a combination of fossil fuel and agricultural emissions, with fluctuations in the $CH_4$ sink in the atmosphere, also plays a part (Schaefer et al., 2016; Worden et al., 2017; Turner et al., 2019; Zhang et al., 2022). The decadal budget indicates that relative uncertainties may range from 20 to 35 % for inventories of anthropogenic emissions in specific sectors (agriculture, waste,

fossil fuels), 50% for emissions from burning biomass and emissions from natural wetland ecosystems, and 100% or more for emissions from other natural sources which include inland waters and geological sources (Saunois et al., 2024). Maasakkers et al. (2023) reported the annual gridded $CH_4$ emission inventory over the United States of America (USA) while meeting the US Environmental Protection Agency (USEPA) emission inventory standards at $0.1° \times 0.1°$ spatial resolution. This data was submitted to the United Nationals in 2020, reporting improved uncertainties over the global Emission Database for Global

Atmospheric Research (EDGAR) database. Geographically, India's wetlands comprise 4.7% of the nation's total land area (Bassi et al., 2014; Kavitha et al., 2016). The primary sources of $CH_4$ emissions include natural emissions from freshwater systems, wetlands, and geological sources; anthropogenic emissions come from waste management, agriculture, and the mining and burning of fossil fuels (Kirschke et al., 2013; Saunois et al., 2016a; Ganesan et al., 2019).

Wetlands are the natural sources that contribute 20 to 40% of global emissions and dominate the inter-annual variability

(Parker et al., 2018). Only limited studies have been conducted in India about $CH_4$ discharge from wetlands. A recent study (Vinna et al., 2021) shows that natural wetlands could produce 50% to 80% more $CH_4$ emissions by 2100. According to Schlesinger et al. (2009), wetlands, rice paddies, and ruminants are the leading producers of $CH_4$ on the Indian sub-continent. According to Hayashida et al. (2013), there is a seasonal pattern in the $CH_4$ concentration over the Indian subcontinent, with higher values during the post-monsoon and minimum in pre-monsoon. Kavitha et al., (2016) used SCIAMACHY retrieved

methane product over the Indian region to understand the spatio-temporal variations. The salient findings of this study are that during monsoon and post-monsoon, high $XCH_4$ values are observed in the Northern regions. Different seasonal behaviour is observed with seasonal peak in post-monsoon and low during monsoon in the Southern peninsular regions. These regional variations are due to the distribution of sources like livestock population, rice cultivation, wetland, biomass burning and oil and gas mining. Along with temperature, precipitation, and radiation, the $CH_4$ emissions from the natural wetlands might affect

the region's heat budgeting, exacerbating global warming on a local, regional, and even global scale (Sakalli et al., 2017). Thermal power plants are responsible for a large amount of the GHG emissions from the energy sector. Each thermal power plant has a different set of emission factors for methane and nitrous oxide, based on operating conditions and combustion technology (Kang et al., 2019). The integrated measure of $CH_4$ includes contributions from the various vertical atmospheric layers, ranging from the Earth's surface measurement point to the uppermost layer of the atmosphere. Chandra et al. (2017)

studied the raised air mass in the 600–200 hPa height layer over northern India, which accounts for 40% of the seasonal $CH_4$ augmentation during the southwest monsoon season. Conversely, in the semi-arid region, the height over 600 hPa contributed up to approximately 88% of the amplitude of the $XCH_4$ seasonal cycle, while the atmosphere below 600 hPa contributed only around 12%. The feature of air mass transport processes in the Asian monsoon region is the main reason for the increased contributions from above 600 hPa across the northern Indian region.

Insufficient datasets exist regarding the $CH_4$ feedback originating from wetlands; a study on the precise estimation of $CH_4$ outflow from wetlands and its impact on local/regional global warming scenarios is urgently needed. The ability to identify spatial and temporal fluctuations in atmospheric $CH_4$ from space, due to recent technological developments in remote sensing, could help fill in the gaps left by measurements performed by ships, planes, and the ground (Frankenberg et al., 2008; Kuze et al., 2009; Kavitha et al., 2018). The present study focuses on the Implications of emissions from coal, thermal power plants, and wetlands on atmospheric methane over South Asia using $XCH_4$ data from Greenhouse gases Observing SATellite (GOSAT) and TROPOspheric Monitoring Instrument onboard the Sentinel-5 Precursor (S5P/TROPOMI) from 2009 to 2022. It has further analyzed the spatial and temporal pattern of atmospheric $CH_4$ variations and emissions and its correlation with anthropogenic $CH_4$ emissions from the bottom-up emission inventory of the EDGAR. The wetland methane emissions were addressed using WetCHARTs v1.31. over the top 10 wetland sites of the present study. The response of atmospheric $CH_4$ concentrations to anthropogenic emissions in various agroclimatic zones of India was further highlighted in this study using the $XCH_4$ data from 2001 to 2022.

## 2  Study Region

The distribution of $CH_4$ sources over the Indian region is shown in Figure 1. The focus of the study was three $CH_4$ source regions—coal fields, thermal power plants, and the Ramsar wetlands. More details about Ramsar wetlands can be found below. The number of coal mines in India varies from 1 to 65, and the top ten coal fields were selected for this study based on their production capacity. During 2019–2020, coal and lignite production ranged between 0.1 and 120.47 MT. The details of the studied coal mines are provided in Table 1. Similarly, Table 2 lists the thermal power stations according to their respective power generation.

The Ramsar Convention is an international agreement created in 1971 to protect wetlands and promote their sustainable use (https://rsis.ramsar.org). As of November 2022, the Ministry of Environment, Forests and Climate Change (MoEF&CC), Government of India, has identified 75 Ramsar Wetland sites in India, spanning a total area of 13,35,530 ha. Based on the total geographical area coverage (Table 3), the top 10 sites were selected for the current investigation. The size of these wetlands ranges from 423,000 ha (Sundarbans Wetland, West Bengal) to 18,900 ha (Wular Lake, Jammu and Kashmir) (https://indianwetlands.in/wetlands-overview/indias-wetlands-of-international-importance/; PIB Press Release on World Wetlands Day dated 26th August, 2022).

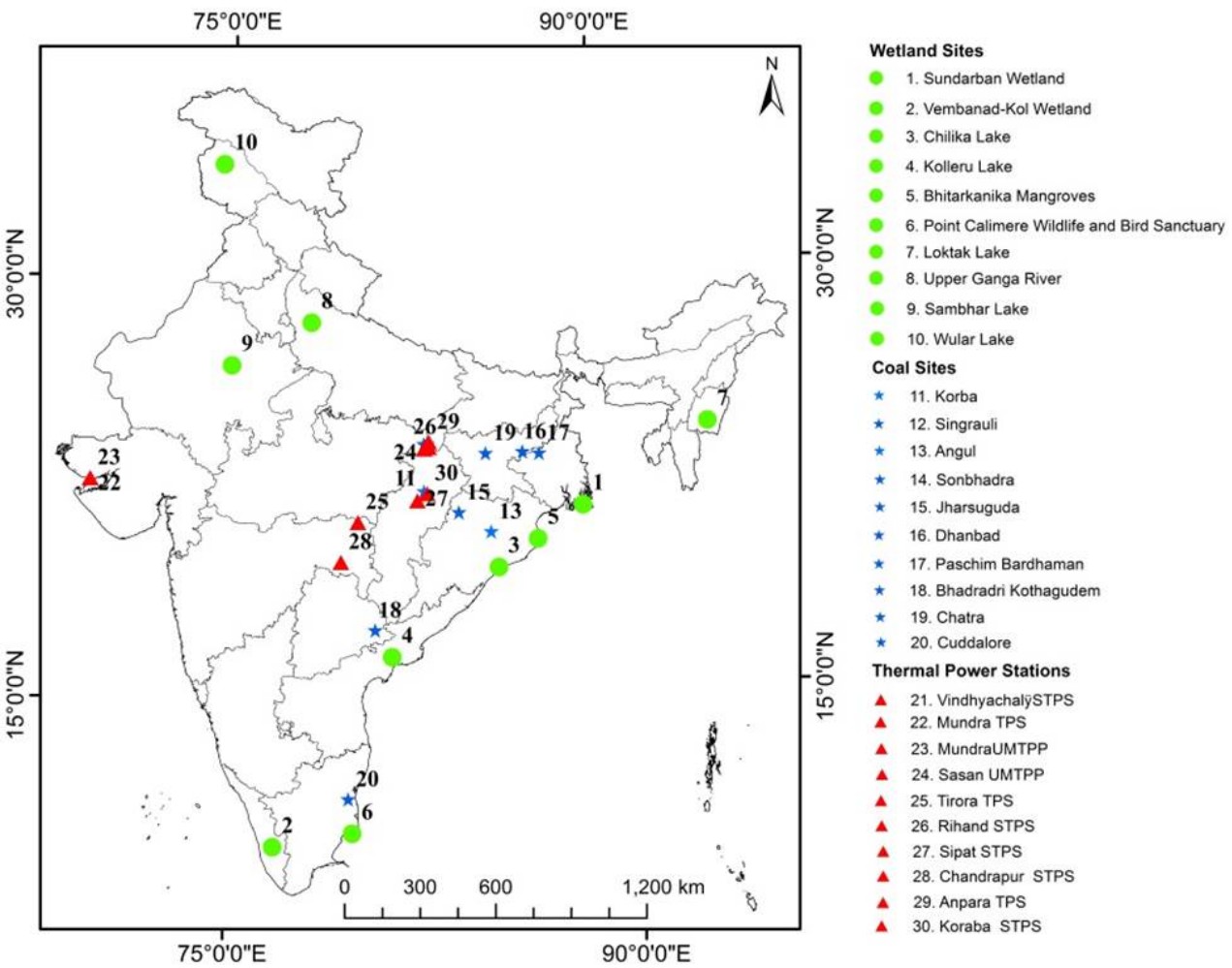

**Figure 1.** Study locations: a) top 10 coal mine locations in India based on production capacity indicated by star (★); b) top 10 thermal power stations denoted by triangle (▲); and c) top 10 wetlands selected based on the area represented by circle (●).

## 3  Data and Methodology

The GOSAT series developed by the Japan Aerospace Exploration Agency (JAXA) continuously monitors $CO_2$ and $CH_4$ from space (Kuze et al., 2009). The present study obtained the level 2 (L2) column $CH_4$ ($XCH_4$) from the GOSAT. Onboard the GOSAT, the Thermal and Near Infrared (NIR) Sensor for Carbon Observation Fourier-Transform Spectrometer (TANSO-FTS) is used to detect the $CO_2$ and $CH_4$ absorption spectra in the shortwave IR ($1.60\mu m$ & $2.0\mu m$) region (Kuze et al., 2009; Kavitha et al., 2018). Ground-based FTIR measurements of $XCH_4$ by the Total Carbon Column Observing Network (TCCON)


are used extensively to validate the GOSAT retrievals. Retrieval bias and precision of column abundance from GOSAT SWIR observations have been estimated as approximately 15-20 ppb and 1%, respectively (Morino et al., 2011; Yoshida et al., 2013). In the present study, the atmospheric $CH_4$ was obtained from 2009 to 2022 within a 100 km radius of the coal mines. The data corresponding to the quality flag (=0) was considered for the study only.

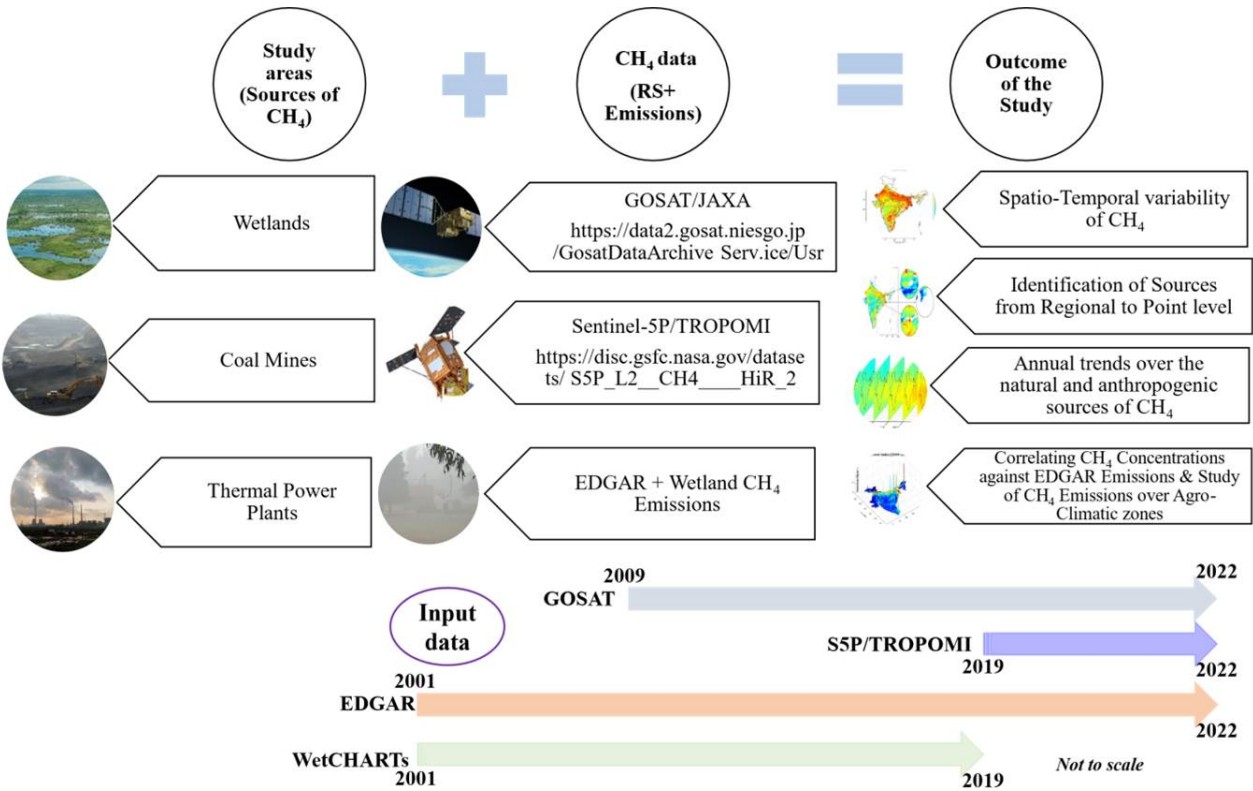

**Figure 2.** Data resources and study approach.

The Sentinel-5 Precursor satellite, launched on October 13, 2017, is equipped with the TROPOspheric Monitoring Instrument (TROPOMI), which tracks cloud characteristics, aerosols, and trace gases (Sentinel-5p, 2019). With a daily pass time of approximately 13:30 local solar time, the instrument's spectrometer measures reflected sunlight in the ultraviolet, visible, NIR, and SWIR spectral windows. The $CH_4$ retrieval algorithm uses the two spectral bands, i.e., reflectance in NIR (757-774 nm) and SWIR (2305-2385 nm) (Kozicka et al., 2023). Initially, retrievals based on TROPOMI had a spatial resolution of 7×7 $km^2$ (along-track × across-track, Lorente et al., 2021). From August 2019 to the present, the resolution has been increased to 5.5×7 $km^2$ (Sagar et al., 2022). The latest data version is now v2 from 2021-07-01 to the present. The quality flag (<0.5) was only considered as per the product readme file document (Sentinel-5p, 2019). Methane retrieval from TROPOMI agrees with ground-based FTIR $XCH_4$ retrievals from TCCON and the Network for Detection of Atmospheric Composition Change

(NDACC). The systematic differences of the bias-corrected XCH$_4$ data with respect to TCCON and NDACC data are, on average, -0.26±0.56% and 0.57±0.83%, respectively (Song et al., 2023).

The data within the coal mines and wetlands area were taken from 1 May 2018 to 30 April 2022. The individual shapefiles were given for each wetland field, and the satellite passes within the area were considered for the current study. As shown in Figure 2, a detailed procedure is explained in this section. The present study utilized the total anthropogenic emissions from

the EDGAR database (https://edgar.jrc.ec.europa.eu/gallery?release=v50&substance=CH4§or=TOTALS, accessed on 1 November 2023). Uncertainties in the information on source intensity, activity, and other statistical data are key parameters for the uncertainties in the EDGAR emission inventory (Janardanan et al., 2017). Bottom-up inventory uncertainties range between 20% and 35% for agriculture, waste, and fossil fuel sectors; 50% for biomass burning and natural wetland emissions; and 100% or higher for natural sources such as geological seeps and inland waters for global methane emissions (Saunois et

al., 2024).

Further, the present study also utilized wetland methane emissions (mg m$^{-2}$ month$^{-1}$) from WetCHARTs v1.3.1 (https://daac.ornl.gov/CMS/guides/MonthlyWetland_CH4_WetCHARTs.html) which is available at a spatial resolution of 0.5°×0.5° with monthly temporal resolution. The scale factor utilized here is 124.5 TgCH$_4$ yr$^{-1}$. We selected the coal fields based on production as shown in Table 1. The data on all coal mines in India, their production, and their location are available in Pai et

al. (2021) and Halder et al. (2024). Each district has open-cast or underground types of mines found in India, and the number of coal mines varies from 1 to 65. The coal/lignite production ranged from 0.1 to 120.47 MT during 2019-2020. The details of the coal mines in the present study are summarized in Table 1, and their locations are mapped in Figure 1.

**Table 1.** The district names, the total number of coal mines, total production, and their centroid (latitudes and longitudes) locations of mines in the respective districts.

| S.No | District names | No. of Mines | Production (MT) | Latitude | Longitude |
|------|----------------|--------------|-----------------|----------|-----------|
| 1 | Korba | 15 | 120.47 | 22.47 | 82.56 |
| 2 | Singrauli | 7 | 82.19 | 24.15 | 82.60 |
| 3 | Angul | 13 | 80.61 | 20.97 | 85.11 |
| 4 | Sonbhadra | 5 | 47.36 | 24.15 | 82.74 |
| 5 | Jharsuguda | 9 | 36.71 | 21.69 | 83.89 |
| 6 | Dhanbad | 51 | 31.25 | 23.76 | 86.46 |
| 7 | Paschim Bardhaman | 65 | 31.23 | 23.68 | 87.11 |
| 8 | Bhadradri Kothagudem | 14 | 30.16 | 17.57 | 80.58 |
| 9 | Chatra | 4 | 29.65 | 23.76 | 85.01 |
| 10 | Cuddalore | 3 | 23.46 | 11.55 | 79.50 |

The list is prepared based on the descending order of total production in each district in India. There are 262 thermal power stations with a full capacity of 229.335 Gigawatt (GW) and a total unit of 2689 in India, based on diesel, gas turbine, and steam as on March 31, 2020. Table 2 shows the list of thermal power stations.

**Table 2.** Top 10 thermal power plants based on their capacity.

| S.No | Power Station Name | Installed Capacity (MW) | No. of Units | Latitude (N) | Longitude (E) |
|------|--------------------|-------------------------|--------------|--------------|---------------|
| 1 | Vindhyachal STPS | 4760 | 13 | 24.1 | 82.68 |
| 2 | Mundra TPS | 4620 | 9 | 22.82 | 69.55 |
| 3 | Mundra UMTPP | 4000 | 5 | 22.82 | 69.53 |
| 4 | Sasan UMTPP | 3960 | 6 | 23.98 | 82.62 |
| 5 | Tirora TPS | 3300 | 5 | 21.41 | 79.97 |
| 6 | Rihand STPS | 3000 | 6 | 24.03 | 82.79 |
| 7 | Sipat STPS | 2980 | 5 | 22.14 | 82.29 |
| 8 | Chandrapur STPS | 2920 | 7 | 20.00 | 79.3 |
| 9 | Anpara TPS | 2630 | 7 | 24.21 | 82.8 |
| 10 | Korba STPS | 2600 | 7 | 22.39 | 82.68 |

There are 11 new Ramsar sites identified in 2022 (total 75 sites) by the Ministry of Environment, Forests and Climate Change (MoEF&CC), India, covering a total area of 1,093,636 ha as of 2022. The present study considered the top 10 sites based on high area coverage (Table 3). The area ranges from 18,900 ha (Wular Lake) to 423,000 ha (Sundarban Wetland).

**Table 3.** Top 10 Wetland fields based on their area coverage.

| S.No | Wetlands Location | Latitude (N) | Longitude (E) | Area (ha) |
|------|-------------------|--------------|---------------|-----------|
| 1 | Sundarban Wetland | 21.77 | 88.71 | 423000 |
| 2 | Vembanad-Kol Wetland | 9.83 | 76.75 | 151250 |
| 3 | Chilika Lake | 19.7 | 85.35 | 116500 |
| 4 | Kolleru Lake | 16.61 | 81.2 | 90100 |
| 5 | Bhitarkanika Mangroves | 20.65 | 86.9 | 65000 |
| 6 | Point Calimere Wildlife Sanctuary | 10.31 | 79.63 | 38500 |
| 7 | Loktak Lake | 24.43 | 93.81 | 26600 |
| 8 | Upper Ganga River | 28.55 | 78.2 | 26590 |
| 9 | Sambhar Lake | 27 | 75 | 24000 |
| 10 | Wular Lake | 34.26 | 74.55 | 18900 |

# 4    Results and Discussion

    ## 4.1    Spatio-temporal variability of Space-based Atmospheric CH$_4$

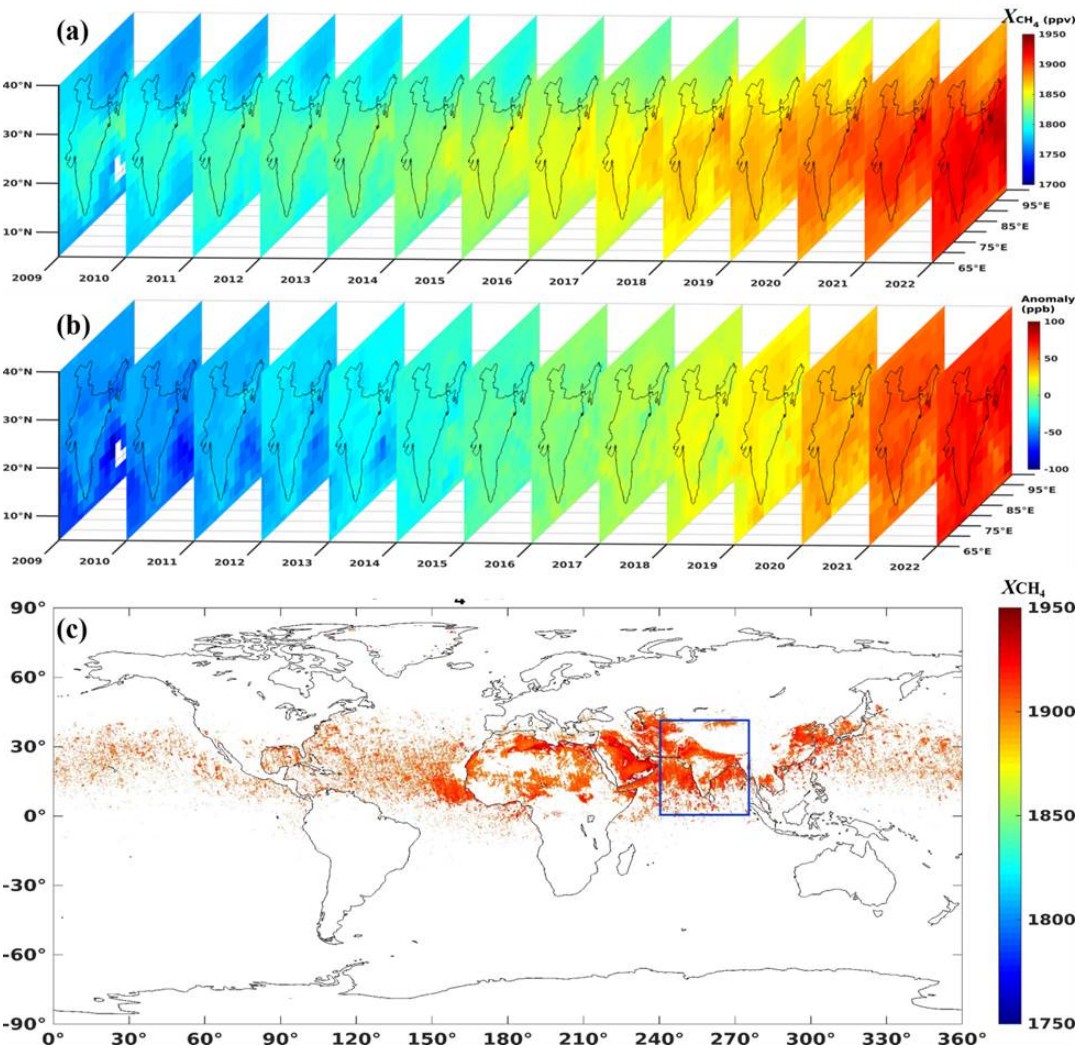

**Figure 3.** a) Remote Sensing (GOSAT) of atmospheric CH$_4$ variability over the Indian sub-continent; b) Anomaly during 2009 to 2022; and c) identification of probable high CH$_4$ concentration using S5P/TROPOMI data from 2019 to 2022 over the study region.

In the present study, we examined the annual space-time distribution of the $X$CH$_4$, obtained from the GOSAT-1 and GOSAT-2 over South Asia as shown in Figures 3a-b from 2009 to 2022 (N=14 years)—the long-term trends in $X$CH$_4$ increased from 1700 ppb to 1950 ppb from 2009 to 2022 with an annual growth rate of 8.76 ppb year$^{-1}$. This growth rate is statistically tested with a p-value less than 0.05 for n=3803 observations. A distinct, evident annual growth in CH$_4$ is seen over the Indian

subcontinent. Figure 3b shows the spatio-temporal residuals calculated using the data from 2009 to 2022. Residuals indicate that the acceleration of $CH_4$ emissions in India has been significant since 2015. Before 2015, the $CH_4$ concentrations were lower by 20 ppb to 50 ppb compared to the total mean of the study period, indicating a slow rise in $CH_4$ activities. However, post-2015, an increase of $CH_4$ was observed at a maximum of 100 ppb compared to the total mean, which indicates the surged emission rates from varied sources of $CH_4$ (Lu et al., 2023). To identify the critical potential high emission zones of $CH_4$, the present study applied the $90^{th}$ percentile statistical filter, as shown in Figure 3c. The percentile is often used to detect the points that are significantly different from the rest of the data. Statistically significant high concentrations of $CH_4$ are observed in tropical regions (Feng et al., 2023). In the blue highlighted box (Latitude: 0°–40°N and Longitude: 60°–100°E), higher concentrations of $CH_4$ were observed in the Indo-Gangetic Plain (IGP) and northwest (NW) areas of India, southeast of China, and NW of China. Southern China and north China are marked with wetlands and rice paddy fields, which are the primary sources of $CH_4$ (Kavitha et al., 2018; Chandra et al., 2019; Guo et al., 2023). High concentrations of $CH_4$ over the IGP and NW of India are due to the population density and various industries that contribute to the emissions of $CH_4$ and emissions from the rice paddy fields, respectively. In the present study, Figure 1 also shows the locations of coal and thermal power plants in India. Globally, the tropical wetlands ecosystem accounts for about 20% of the total global source (Saunois et al., 2020; Shaw et al., 2022), evidenced by bottom-up and top-down inventories. The study in the following sections assessed the $CH_4$ growth rate associated with the source type over the Indian region.

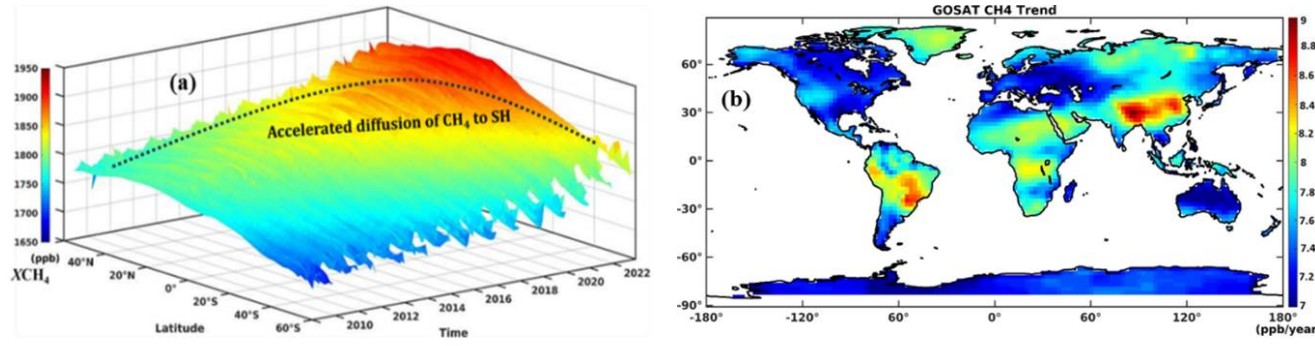

**Figure 4.** (a) Spatiotemporal distribution of annual $XCH_4$ as a function of latitude during 2010 to 2022. (b) Global $XCH_4$ trend (ppb year$^{-1}$) using GOSAT data.

Figure 4 shows the spatiotemporal distribution of $XCH_4$ as a function of latitude, which depicts the annual variability at each latitude covering the northern and southern hemispheres (SH). There is a transparent latitudinal gradient in space. A strong diffusion of $CH_4$ is observed from the northern hemisphere to SH during 2009 to 2022. During 2010, the $XCH_4$ was distributed nearly constantly at all latitudes, indicating the stability of emissions from natural and anthropogenic sources. However, the gradient between the NH and SH has narrowed down with a growth rate of 12 ppb year$^{-1}$ in 2022, reflecting the dominance of anthropogenic emissions over the tropics and unidentified leaks from the tropical wetlands and natural gas (Rocher-Ros et al., 2023). More thoroughly, the characteristics of regional and global spatiotemporal variations are revealed by the continuous

$X$CH$_4$ data in space and time. As shown in Figure 4, it displays a latitudinal gradient, and each latitudinal zone's growth tendencies are comparable. Figure 4b shows $X$CH$_4$ has increased the global mean trend from 7 ppb year$^{-1}$ to 9 ppb year$^{-1}$. There are hotspots in the $X$CH$_4$ trend observed in the Tibetan plateau (8.2 to 9 ppb year$^{-1}$), South America (8.2 to 8.8 ppb year$^{-1}$), and the African continent (8 to 8.4 ppb year$^{-1}$), and in the rest of the world varies from 6.75 to 8 ppb year$^{-1}$. Similarly high values were reported in the Tibetan plateau from 2010 to 2022 (Wei et al., 2019) and 8 ppb year$^{-1}$ (Song et al., 2023) from 2009 to 2021 using GOSAT data.

### 4.2   Assessment of $X$CH$_4$ over different source types in India

Figures 5a-c show the monthly time series of $X$CH$_4$ over the specific sources of CH$_4$ plotted in the Indian region from 2009 to 2022. Over the Indian sub-continent and south-east Asia, October to November exhibits the highest amounts of CH$_4$, while March through June often sees the lowest (Sreenivas et al., 2016; Song et al., 2023), because of the enormous diversity in the climate zones of the Asian region. The seasonal cycle (peak and trough) of $X$CH$_4$ is strongly associated with the vegetation during the active phase of cultivation and reduced photochemical reaction by the hydroxyl radicals, respectively. The major sink for CH$_4$ is by the oxidation of OH radical in the troposphere which removes 90% of CH$_4$ from the atmosphere (Crutzen et al., 1991). However, the potential available of OH radical in the atmosphere is not steady and changes rapidly depending upon the presence of solar ultraviolet radiation and other trace gases such as ozone, oxides of nitrogen (NO+NO$_2$), and water vapor (Sreenivas et al., 2016).

Over the coal, thermal, and wetlands, the $X$CH$_4$ shows typical seasonal behavior, with maximum activity during the post-monsoon (October-November) and minimum activity in the pre-monsoon (March-May), as shown in Figure 6. A seasonal maximum of $X$CH$_4$ was observed over coal and thermal power plants from September to October and a minimum in pre-monsoon (March to May). In the case of wetlands, a shift in seasonal maxima varies from site to site, indicating their respective active phase of methanogens and the magnitude of the seasonal amplitude, which runs as a function of the individual wetland area. Methanogens are microscopic organisms that break down organic substances in an oxygen-free environment. Thus, wetlands are perfect for methanogens to grow and release CH$_4$ since they are usually oxygen-poor, moist habitats (Zhang et al., 2023). Therefore, the present study investigated the above-listed wetlands. Most of the wetlands exhibit an annual growth rate of $X$CH$_4$ greater than 9.50 ppb year$^{-1}$ with high concentrations over Sundarbans wetland (Area = 423000 ha) with pronounced seasonality at all sites and lower concentrations over Wular Lake (area = 18900 ha) with an annual trend of 8.72$\pm$0.3 ppb year$^{-1}$. During the examination period, the seasonal trends (slope) at each location, as summarised in Tables 1-3, were evaluated using Sen's slope-based Mann-Kendall test with a significance of p-value < 0.05 (Pathakoti et al., 2021). The rate of increase in $X$CH$_4$ was higher over the Upper Ganga (area = 26590 ha) with a slope of 9.82$\pm$0.52 ppb year$^{-1}$ and followed by Vembanad-Kol Wetland (area = 151250 ha) with a slope of 9.69$\pm$0.44 ppb year$^{-1}$. Over the Sundarbans wetlands, West Bengal (area= 423000 ha) the rate of increase in $X$CH$_4$ is 9.54$\pm$0.51 ppb year$^{-1}$. To investigate further, the present study quantified the source-based natural CH$_4$ fluxes from each wetland using the WetCHARTs data in the following section.

Typically, the Indian climate is hot and humid, causing disturbances in the rainfall patterns; an increase in the waterlogged soils expands the wetlands (Zhang et al., 2023). Typical tropical wetlands are acting as positive feedback to climate change

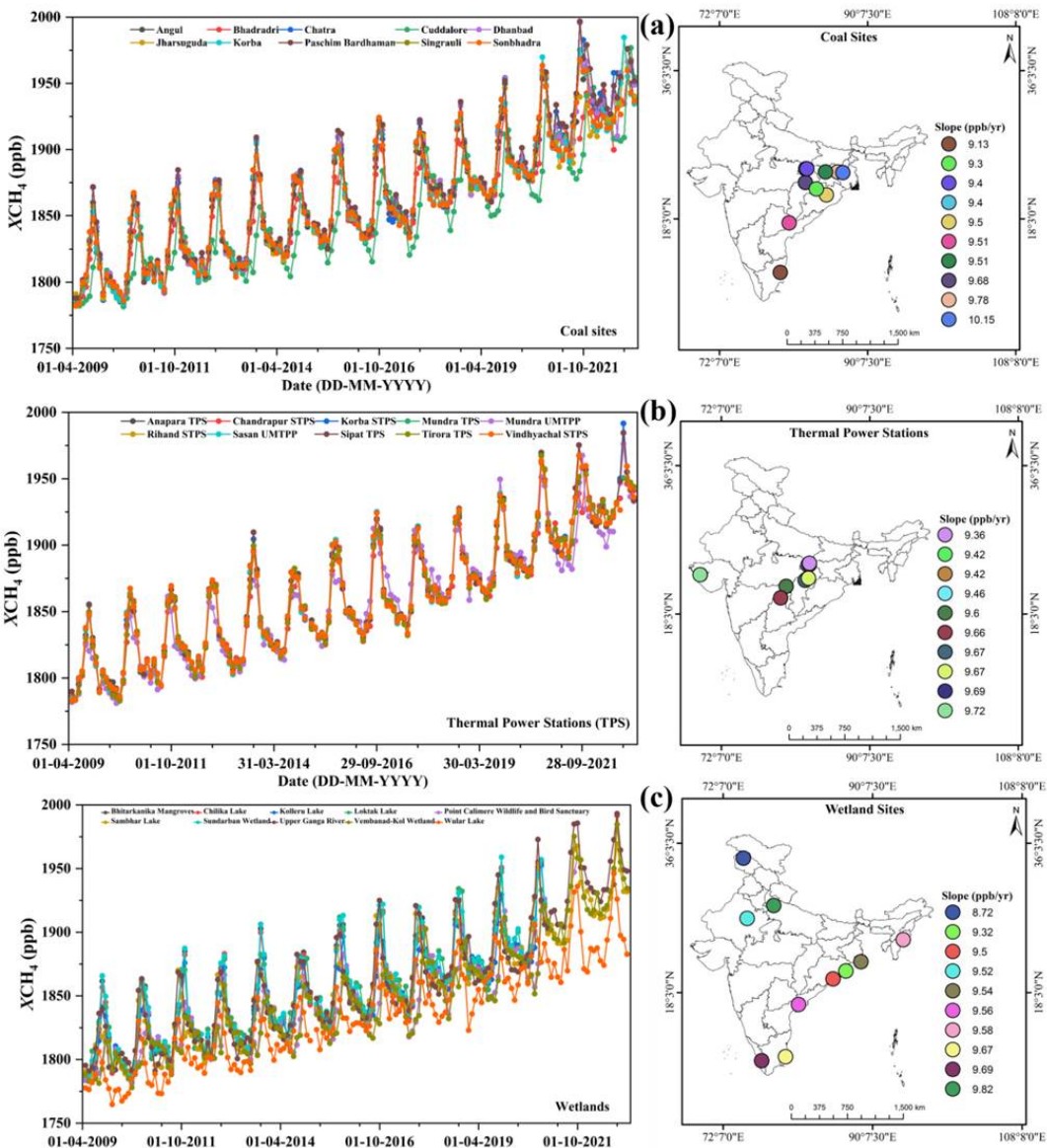

**Figure 5.** Monthly time series of $X$CH$_4$ over the a) wetlands, b) thermal power stations, and c) coal fields: sources of emissions, along with the overall growth rate at the respective site.

(Salimi et al., 2021). Irrespective of the power production capacity, over the thermal power plants, the CH$_4$ exhibited stabilized seasonality at each location. However, the growth rate of $X$CH$_4$ was higher over the Mundra Ultra Mega power plant (UMPP), Gujarat with a slope of 9.72±0.41 ppb year$^{-1}$ followed by Mundra Thermal power station with a slope of 9.69±0.4 ppb year$^{-1}$.

195 The Mundra TPS and UMPP, Gujarat have a total power capacity of 8620MW with 14 units. With 2630 MW installed power

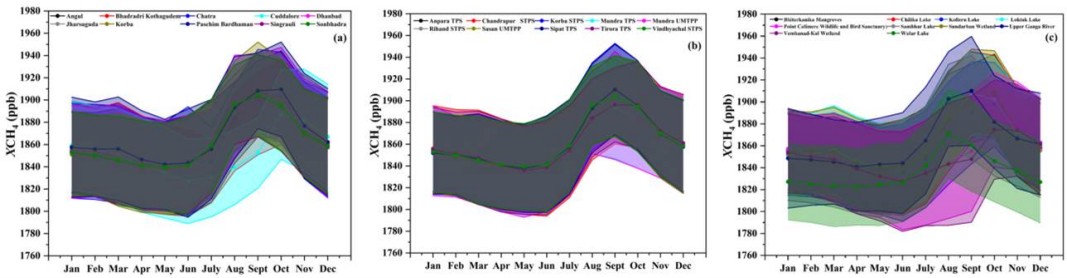

**Figure 6.** Seasonal $XCH_4$ over a) coal fields, b) thermal power stations, and c) wetlands.

capacity the Anpara TPS exhibited an $XCH_4$ growth rate of 9.36$\pm$0.5 ppb year$^{-1}$. This indicated the higher potential power plants contribute more $CH_4$ emissions into the atmosphere. Over the coal mines, Paschim Bardhman (31.23 MT, 65mines) shows a high $XCH_4$ trend of about 10.15$\pm$0.55 ppb year$^{-1}$ followed by Dhanbad (31.25 MT, 51mines), Korba (120.47 MT, 15mines) which shows $XCH_4$ trend of 9.78$\pm$0.53 ppb year$^{-1}$ and 9.68$\pm$0.52 ppb year$^{-1}$, respectively. Angul (80.61MT, 13 mines) and Chatra (29.65 MT, 4 mines) show $XCH_4$ trend of 9.51$\pm$0.5 ppb year$^{-1}$. The lowest annual trend in $XCH_4$ was observed over the Cuddalore coal mine (23.46 MT, 3 mines) which is about 9.13$\pm$0.4 ppb year$^{-1}$. Anthropogenic emissions influence the methane growth trend. Wetland and biomass burning emissions determine the interannual variability (Fo et al., 2024).

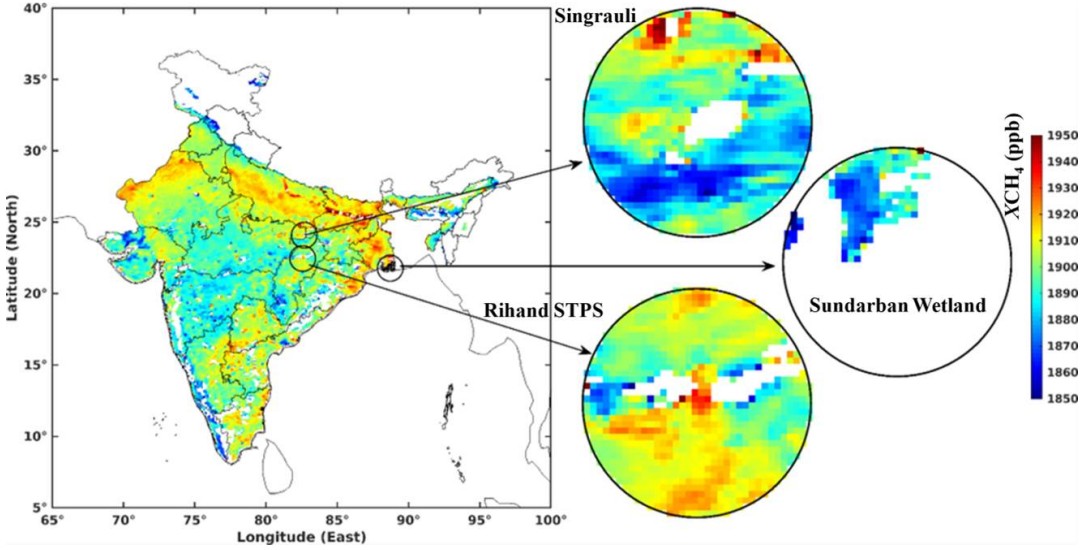

**Figure 7.** S5P/TROPOMI $XCH_4$ gridded to $0.05° \times 0.05°$ over the Indian region and $XCH_4$ over wetland, coal, and thermal power plant sites with a radius of 100 km.

Figure 7 shows the continuous $XCH_4$ data from the S5P/TROPOMI at $0.05° \times 0.05°$, complementing the GOSAT efforts in monitoring the $XCH_4$ dynamics in space and time. We demonstrated the spatiotemporal variation characteristics of $XCH_4$ more

comprehensively at three different source type locations (wetland, coal, and thermal power plant). High $XCH_4$ concentrations over the coal and thermal power station sites, and relatively lower concentrations in the wetland site. We concluded that the high-resolution S5P/TROPOMI has the potential to detect the point source variability. The growth rates of $XCH_4$ over the wetlands compete with coal sites, indicating an equivalent anthropogenic source. Results of the analysis in the context of thermal power plants and coal mines indicate that the emissions from the fossil fuel industries are significant, and the release of $CH_4$ into the atmosphere is commensurate with the production of the power and mining capacity.

### 4.3 CH$_4$ emission from India's wetlands

In addition to the anthropogenic emissions, the present study utilised the global monthly wetlands emission estimates from the Wetland Methane Emissions and Uncertainty (WetCHARTs v1.3.1) inventory (Bloom et al., 2017a, b; Bloom et al., 2021 ). The figure 8a shows the monthly $CH_4$ emission over India's top 10 wetland sites from 2001 to 2019 and figure 8b represents the long-term seasonally averaged $CH_4$ emission over wetlands. Emissions of $CH_4$ from the wetlands were minimal in the winter season (December to February) and pre-monsoon (March to May). In the tropical region, winter and pre-monsoon seasons are considered dry months with moderate to high temperatures and less precipitation.

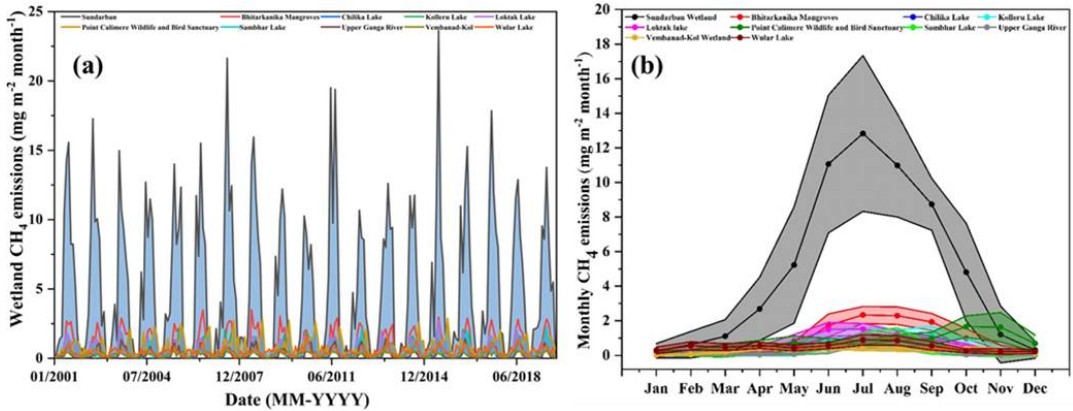

**Figure 8.** (a) Monthly Time series of Methane emissions (mg m$^{-2}$ month$^{-1}$) over the wetland sites, (b) Seasonal methane emissions over the wetland sites from 2001 to 2019.

A study by Peng et al. (2022) and Feng et al. (2022) hypothesized that warmer and wetter wetlands contribute significantly to the high $CH_4$ emissions to the atmosphere. Typical climatological (1991-2020) mean temperature (accumulated seasonal precipitation) over India during winter, pre-monsoon, monsoon, and post-monsoon are 20 °C (23 mm), 28 °C (98 mm), 26 °C (867 mm) and 23 °C (106 mm) respectively (https://climateknowledgeportal.worldbank.org/country/india/climate-data-historical). At all the wetland study sites during the study period, the maximum $CH_4$ emission was reported during the monsoon months with a maximum value of 23.62±3.66 mg m$^{-2}$ month$^{-1}$ over the Sundarbans wetland, which is the largest protected wetland of India and mangrove forest in the world. Besides climatic conditions, the emissions of $CH_4$ are positively correlated

with the size of the wetland, thus reporting maximum $CH_4$ emission over the Sundarbans site. High natural $CH_4$ emissions during the monsoon positively correlate with the atmospheric $XCH_4$ concentrations.

Further, Mann-Kendall-based statistical analysis was carried out to assess the annual trend in the $CH_4$ emissions and found significant trend over the Wular Lake, with an increasing rate of 0.04 mg m$^{-2}$ year$^{-1}$ with a p-value of 0.01. An annual trend of $XCH_4$ was over this study is about 8.72±0.3 ppb year$^{-1}$. The current research highlights the need for further investigation to correlate in detail the temperature and associated precipitation influence on methane oxidation and microbial activities, thus modulating the $CH_4$ emissions from the wetlands.

## 4.4 Long term seasonal winds over the source locations

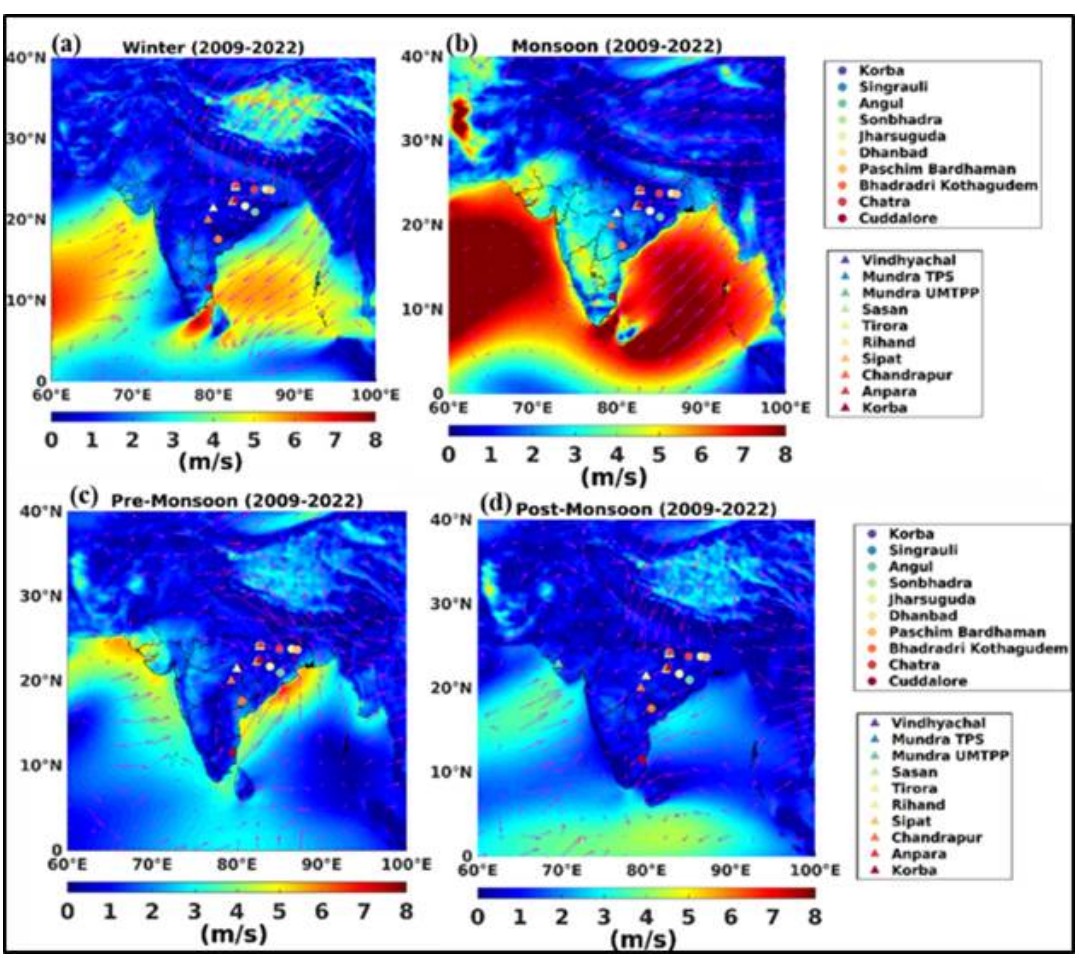

**Figure 9.** Long-term seasonal winds for source types (coal and thermal power plants) for the period 2009-2022: (a) winter, (b) monsoon, (c) pre-monsoon, and (d) post-monsoon respectively.

The 10 m u wind component of wind from ERA5 which is reanalysis product is used in the present study for the period 2009-

2022. The Figure 9 shows the long-term seasonal winds for coal and thermal power plants for the period 2009-2022. During winter, winds are primarily from northeast (NE) direction with low wind speeds at the study locations. During monsoon and pre-monsoon majority of winds are from southwest (SW) direction with medium to high wind speeds arriving at the source locations. At the coal and thermal power plants, winds from the SW during the pre-monsoon, which transports relatively clean airmass from the ocean to land, could also influence the observed low $CH_4$ concentrations along with the continued source

activity and seasonality. Quantification of $CH_4$ fluxes with an improved accuracy also needs required information about the prevailing winds at the source locations. Bussmann et al. (2024) established a detailed relationship between modulation in $CH_4$ fluxes against the wind speed and direction in his studies.

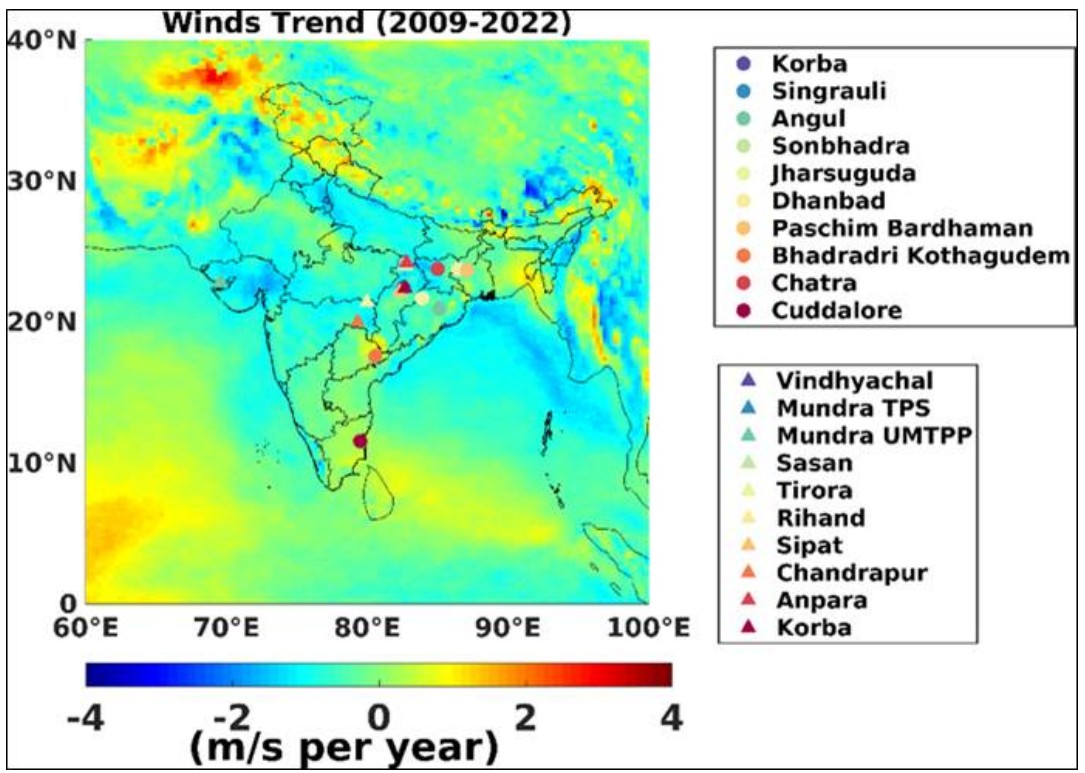

**Figure 10.** Long-term spatial trend of winds over the Indian region covering source types (coal and thermal power plants) for the period 2009-2022.

Figure 10 shows the trend in wind speeds over the coal and thermal power plants from 2009-2022. Around the coal mine sites, a positive trend in wind speed is observed over the Cuddalore coal mine site (0.42 ms$^{-1}$ year$^{-1}$), whereas the remaining

coal mine sites show a negative trend in wind speed. The maximum negative trend is observed over the Sonbhadra coal mine site (1.21 ms$^{-1}$ year$^{-1}$), and the minimum negative trend is observed over the Bhadradri Kothagudem coal mine (0.18 ms$^{-1}$ year$^{-1}$). A negative trend in wind speed is observed over all the thermal power plants, with a maximum trend observed over

the Vindhyachal STPS ($1.23 \ \mathrm{ms}^{-1} \ \mathrm{year}^{-1}$) and a minimum over the Sipat STPS ($0.63 \ \mathrm{ms}^{-1} \ \mathrm{year}^{-1}$). Over the wetland sites, a positive trend in wind speed is observed over the Point Calimere Wildlife and Bird Sanctuary ($0.25 \ \mathrm{ms}^{-1} \ \mathrm{year}^{-1}$) and Wular

Lake ($0.20 \ \mathrm{ms}^{-1} \ \mathrm{year}^{-1}$). All the remaining wetland sites show a negative trend in wind speed. Besides the surface emissions, column $CH_4$ values are also varied by the background flow advection and unstable boundary layer due to strong convection (vertical mixing) in the daytime (Ricaud et al., 2014; Francis et al., 2023). A negative trend is observed over the source locations in the present study, indicating relatively slower dispersion at these locations, thus modulating column $CH_4$ values.

### 4.5   $CH_4$ emissions over India's Agroclimatic zones

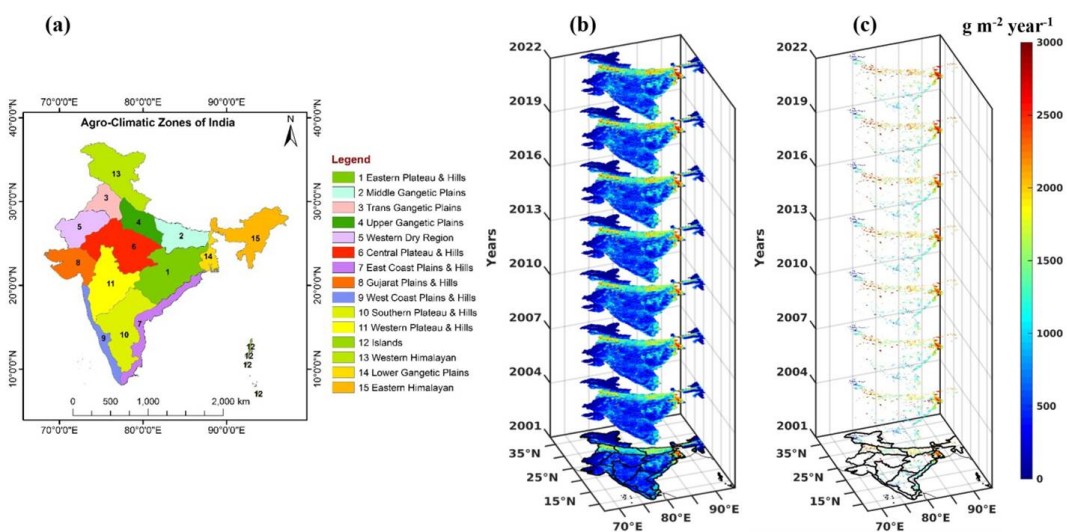

**Figure 11.** a) Agroclimatic zones of India, b) bottom-up $CH_4$ emissions inventory of EDGAR, and c) $90^{th}$ percentile statistical filter applied on $CH_4$ emissions from 2001 to 2022.

India is divided into 15 agroclimatic zones according to the combination of soil types and climatic conditions (Choudhary and Sirohi, 2022). These zones offer a structure for the nation's development and execution of agricultural policies and practices. The crops and farming methods that are most appropriate for the environmental conditions in each zone are distinct from one another. Out of natural and anthropogenic sources of $CH_4$, agricultural activity is also one of the dominant contributors to $CH_4$ dynamics in the atmosphere. Figures 11a-c show India's 15 agroclimatic zones and spatiotemporal trends of $CH_4$

emissions obtained from the bottom-up emission inventory of EDGAR (Crippa et al., 2020) from 2001 to 2022. Significant high emissions of $CH_4$, as shown in Figure 11c, were reported over the Middle Gangetic Plains-MGP (2), Trans Gangetic Plains-TGP (3), Upper Gangetic Plains-UGP (4), East Coast Plains & Hills-ECPH (7), Lower Gangetic Plains-LGP (14) and East Gangetic Plains-EGP (15). These agroclimatic zones have active farming in rice, wheat, sugarcane, maize, millet, gram, cotton, etc. Besides traditional farming, the Lower Gangetic Plains has also actively contributed to livestock, horticulture,

and forage production (Ahmad et al., 2017). Among all 15 agroclimatic zones, the MGP, TGP, UGP, ECPH, LGP and EGP

have exhibited high emissions of CH$_4$ indicating the diversification of agricultural practices and homogenous traditions of agricultural production. Rice- wheat (R-W) based production system is mainly being practiced in this region which is causing negative effects on climate (Taneja et al., 2019). CH$_4$ emissions over the Northwest region are exhibiting weak contribution compared to other agroclimatic zones of India.

### 4.6    Spatial correlation between $X$CH$_4$ concentrations and emissions over India

To understand the relationship between India's high $X$CH$_4$ concentration zones against emissions, we have computed pixel-level correlation between S5P/TROPOMI measured $X$CH$_4$ concentrations and bottom-up inventory of EDGAR-based $X$CH$_4$ anthropogenic emissions. Figures 12a-c shows $X$CH$_4$ concentrations from S5P/TROPOMI, EDGAR-based anthropogenic emissions, and their correlation coefficient. The spatial patterns of $X$CH$_4$ concentrations agree well with the high-emission regions. The correlation coefficient 'r' is strongly positive on in the IGP region, shows that more CH$_4$ emission into the atmosphere through rapid industrial activity and anthropogenic contribution from human activity due to high population density. Besides the IGP region, the 'r' value is also strong in the east and northeast region due to the emissions from natural sources such as agricultural activities, livestock, and wetlands (Behera et al., 2022).

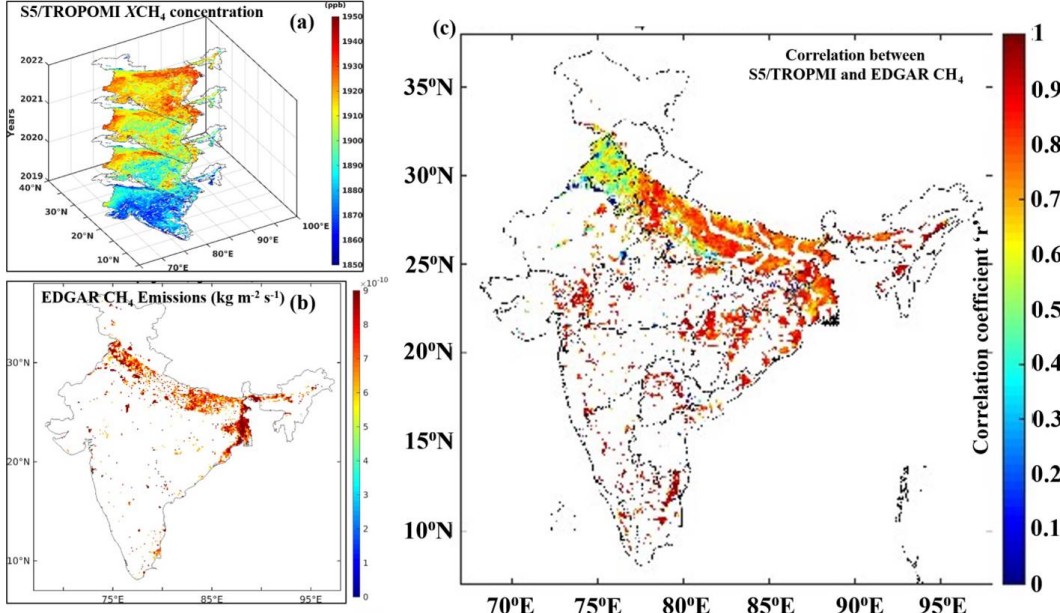

**Figure 12.** Pixel level correlation between S5P/TROPOMI based a) $X$CH$_4$ concentrations, b) anthropogenic CH$_4$ bottom-up emission inventory of EDGAR during 2019 to 2022, and c) correlation map.

## 5   Conclusions

Since the beginning of the Industrial Revolution, growing human populations have resulted in increased waste production, agriculture, and the use of fossil fuels. Therefore, this study demonstrated the spatiotemporal dynamics of $X\mathrm{CH}_4$ in the atmosphere and associated natural (wetlands) and anthropogenic sources (coal fields and thermal power plants) in the Indian region. The present study utilized the remote sensing based $X\mathrm{CH}_4$ data from the GOSAT and S5P/TROPOMI from 2009 to 2022. The following are the salient findings of the study.

– The present study demonstrated the continuous $X\mathrm{CH}_4$ data from the S5P/TROPOMI and GOSAT to effectively monitor the $X\mathrm{CH}_4$ dynamics in space and time.

– Long-term trends of $X\mathrm{CH}_4$ show significant annual growth from 2009 to 2022 in $\mathrm{CH}_4$ over the Indian subcontinent, with a yearly growth rate of 8.76 ppb which is in line with the global trend.

– Long-term temporal and spatial distribution characteristics and variations of $\mathrm{CH}_4$ emissions in India have accelerated in
the last decade and globally, a substantial diffusion of $\mathrm{CH}_4$ is observed from the northern to southern hemisphere.

– $X\mathrm{CH}_4$ levels peak in September-October over coal and thermal power plants but reach their minimum during March-May. The seasonal maxima of wetlands vary from site to site and are related to their size and active phase of methanogens.

– Majority of the wetlands show an annual growth rate in $X\mathrm{CH}_4$ is about 9.50 ppb year$^{-1}$, indicates rich in moist habitats and active methanogens process.

– High $X\mathrm{CH}_4$ trend of 9.72±0.41 ppb year$^{-1}$ from the Mundra UMPP, Gujarat as well as the Paschim Bardhman coal mine (slope of 10.15±0.55 ppb year$^{-1}$) indicated elevated and significant emissions from fossil fuel industries as compared to other natural sources.

– The highest $\mathrm{CH}_4$ emission estimate was observed during the monsoon season over the Sundarbans wetland, the largest protected wetland in India, with a maximum value of 23.62±3.66 mg m$^{-2}$ month$^{-1}$. Among the wetland sites, Wular
Lake has a rising methane rate of 0.04 mg m$^{-2}$ month$^{-1}$ with a p-value of 0.01.

– The high levels of $\mathrm{CH}_4$ emissions seen in the MGP, TGP, UGP, ECPH, LGP, and EGP agroclimatic zones may be related to the varied farming methods and traditional agricultural output in these regions. Most of these areas revolve around the Rice-Wheat farming system which is negatively impacting the climate.

– The spatial patterns of $X\mathrm{CH}_4$ concentrations agree well with the high-emission regions. The correlation coefficient 'r' is
strongly agreed in the IGP region.

– Therefore we conclude that the space based $X\mathrm{CH}_4$ dataset provides significant support to track long-term changes in $\mathrm{CH}_4$ and provides insightful information on the causes and feedback mechanisms for the elevated concentrations of methane across the south Asia region.

*Code availability.* The code used in the present study will be available from the author upon request.

*Data availability.* GOSAT (https://data2.gosat.niesgo.jp/GosatDataArchiveServ.ice/Usr), TROPOMI (https://disc.gsfc.nasa.gov/datasets/S5P_L2_CH4_HiR_2), EDGAR bottom-up inventory (https://edgar.jrc.ec.europa.eu/gallery?release=v50&substance=CH4§or=TOTALS), and wetland methane emissions and uncertainty data (https://daac.ornl.gov/CMS/guides/MonthlyWetland_CH4_WetCHARTs.html) used in the present study are freely available and can be downloaded as summarized in Figure 2 with the user's credentials.

*Author contributions.* D.V. Mahalakshmi: Conceptualization, Formal analysis, Writing – original draft. Mahesh P: Conceptualization, For-
mal analysis, Writing – original draft. A.L. Kanchana: Conceptualization, Formal analysis, Writing – original draft. Sujatha P: Formal data analysis, Writing – original draft. Ibrahim Shaik: Analysis. K.S. Rajan: Writing – review and editing. Vijay Kumar Sagar: Formal analysis and data curation. P. Raja: Writing – review and editing. Y.K. Tiwari: Writing – review and editing. Prakash Chauhan: Writing – review and editing.

*Competing interests.* The authors declare that they have no conflict of interest.

*Acknowledgements.* We sincerely thank the Director, NRSC-ISRO, for his kind guidance and support. Authors greatly acknowledge the JAXA and National Institute of Environmental Studies (NIES) for providing free access to the GOSAT $X$CH$_4$ observations (https://data2.gosat.niesgo.jp/GosatDataArchiveServ.ice/usr/download, accessed on 15 June 2023) and the Earthdata for giving access to the S5P/TROPOMI (https://disc.gsfc.nasa.gov/datasets/S5P_L2_CH4_HiR_2/summary, accessed on 15 June 2023) data. We also thank the European Commission's Joint Research Centre (JRC) for providing the CH$_4$ bottom-up inventory of EDGAR (https://edgar.jrc.ec.europa.eu/gallery?release=
v50&substance=CH4§or=TOTALS, accessed on 25 July 2023). The authors also thank the Oak Ridge National Laboratory (ORNL) Distributed Active Archive Center (DAAC) for providing the wetland methane emissions data. This work has been carried out as part of the Technology Development Project (TDP) titled "Investigation of Atmospheric GHGs Emissions over the Indian Region (AGE)" and the Land-Ocean-Atmospheric GHGs Interaction Experiments (LOAGIN-X) of Climate and Atmospheric Processes of ISRO-Geosphere Biosphere programme (CAP-IGBP). We are very grateful to the anonymous reviewers and the handling editor for their constructive comments
and suggestions, which have helped us to improve the manuscript.

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
