# Peer review of "Emissions of methane from coal thermal power plants and wetlands and its implications on atmospheric methane across the South Asian region"

_EGUsphere, 2024_

## Referee Comment (RC2)

The current manuscript titled "**Emissions of Methane from Coal, Thermal power plants and Wetlands and its implications on Atmospheric Methane across the South Asian Region**" by Mahalakshmi et al., carried out a detailed study on atmospheric column CH4 using the satellite data and bottom-up emission inventory data. This work is well executed and importantly carried out an extensive analysis over different source type of methane which is an important approach. The content is well-written and structured. The study looked into the effects of changes Emissions of Methane from Coal, Thermal power plants and Wetlands and its implications on Atmospheric Methane across the South Asian Region. Author (s) could use potentially the S5P/TROPOMI observations to map the point level sources. The present study has many important points in which it highlights the emission source versus concentrations in the IGP region. Also they carried out methane emissions variability in the agroclimatic zones.

Therefore, I believe this paper may be accepted with the following minor corrections.

1. In the title capitalisation of every letter of the word may not be required.
2. 205: Is there any references supporting this statement "higher concentrations of CH4 were observed in the Indo-Gangetic Plain (IGP) and northwest (NW) areas of India, southeast of China, and NW of China. Southern China and north China are marked with wetlands and rice paddy fields, which are the primary sources of CH4"
3. Figures 5c adjust the x axis scale accordingly with the Fig. 5(a) and (b)
4. There is a typo in the caption of Figure 7, indi. "Figure 7. S5P/TROPOMI XCH4 gridded to 0.05° × 0.05° over Indi and XCH4 over wetland, coal, and thermal power plant sites with a radius of 100 km"
5. Significant high emissions of CH4, as shown in Figure 7c, but there is no Figure 7c it is missing.
6. Figures 7 and 8 are described differently than their respective figures. Furthermore, the description of Figure 8 comes before that of Figure 7.

---

## Author Comment (AC2)

**Response to Reviewer-2**

The current manuscript titled "**Emissions of Methane from Coal, Thermal power plants and Wetlands and its implications on Atmospheric Methane across the South Asian Region**" by Mahalakshmi et al., carried out a detailed study on atmospheric column $CH_4$ using the satellite data and bottom-up emission inventory data. This work is well executed and importantly carried out an extensive analysis over different source type of methane which is an important approach. The content is well-written and structured. The study looked into the effects of changes Emissions of Methane from Coal, Thermal power plants and Wetlands and its implications on Atmospheric Methane across the South Asian Region. Author (s) could use potentially the S5P/TROPOMI observations to map the point level sources. The present study has many important points in which it highlights the emission source versus concentrations in the IGP region. Also they carried out methane emissions variability in the agroclimatic zones. Therefore, I believe this paper may be accepted with the following minor corrections.

Reply: Thank you for the overall comments and positive feedback provided from the present study. All the comments provided by you are addressed in the revised manuscript line-by-line.

1. In the title capitalisation of every letter of the word may not be required.

Reply: Thank you for the suggestion. The title is modified as "Emissions of methane from coal, thermal power plants and wetlands and its implications on atmospheric methane across the South Asian region" and same has been updated in the revised manuscript.

2. 205: Is there any references supporting this statement "higher concentrations of $CH_4$ were observed in the Indo-Gangetic Plain (IGP) and northwest (NW) areas of India, southeast of China, and NW of China. Southern China and north China are marked with wetlands and rice paddy fields, which are the primary sources of $CH_4$"

Reply: Thank you for the suggestion. The references supporting the above statement are (Kavitha et al., 2018; Chandra et al., 2019; Guo et al., 2023).The same has been updated in the revised manuscript.

3. Figures 5c adjust the x axis scale accordingly with the Fig. 5(a) and (b).

Reply: Figures 5a-c shows the monthly time series of $XCH_4$ over the specific sources of $CH_4$ plotted in the Indian region during 2009 to 2022 along with the overall growth rate at the respective site from 2009 to 2022. The revised figure is updated with 2021 and 2022 data and x axis scale is made uniform for the figures 5a,b,c.

4. There is a typo in the caption of Figure 7, indi. "Figure 7. S5P/TROPOMI XCH4 gridded to $0.05° \times 0.05°$ over Indi and XCH4 over wetland, coal, and thermal power plant sites with a radius of 100 km"

Reply: The typo in the caption of Figure 7 is corrected as below

Figure 7.S5P/TROPOMI $XCH_4$ gridded to $0.05° \times 0.05°$ over Indian region and $XCH_4$ over wetland, coal, and thermal power plant sites with a radius of 100 km.

5. Significant high emissions of $CH_4$, as shown in Figure 7c, but there is no Figure 7c it is missing.

Reply: The figure in section 4.3 is figure 9c instead of figure 7c and the same has been corrected in the revised manuscript.

6. Figures 7 and 8 are described differently than their respective figures. Furthermore, the description of Figure 8 comes before that of Figure 7.

Reply: Thank you for the comment. The figures numbers were wrongly written in the text of the manuscript. Figure 7 shows the continuous $X$CH$_4$ data from S5P/ TROPOMI complementing the GOSAT efforts instead of figure 8 which was written in the manuscript. In section 4.3, significant high emissions of $CH_4$ as shown in figure 9c, instead of figure 7c which was written in the manuscript. The figures numbers were corrected in the revised manuscript.

---

## Author Comment (AC3)

**Reviewer -1**

**Response to Reviewer-1**

The manuscript "Emissions of Methane from Coal, Thermal power plants and Wetlands and its Implications on Atmospheric Methane across the South Asian Region" investigates the spatial and temporal dynamics of atmospheric methane ($CH_4$) concentrations over the South Asia region from 2009 to 2022, utilizing data from the Greenhouse gases Observing SATellite (GOSAT) and the TROPOspheric Monitoring Instrument onboard the Sentinel-5 Precursor (S5P/TROPOMI). The analysis identifies specific sources contributing to this increase, notably the Mundra thermal power station and Mundra ultra mega power plant, exhibiting higher rates of increase in $XCH_4$ compared to other natural and anthropogenic sources. The study also highlights the significant methane emissions from the Sundarbans natural wetland, competing with coal sites in terms of emission rates, thus emphasizing its importance as an equivalent anthropogenic source. Furthermore, the investigation delves into the distribution of $CH_4$ emissions across 15 Indian Agroclimatic zones; and employs bottom-up anthropogenic $CH_4$ emissions data to map against $XCH_4$ concentrations, revealing a high correlation in the Indo Gangetic Plains (IGP) region, thereby identifying key anthropogenic $CH_4$ hotspots. Overall, the manuscript provides crucial insights into the impact of both natural and anthropogenic sources on $XCH_4$ concentrations over the Indian region, quantifying spatio-temporal changes at each study site. The findings hold significance for understanding and addressing the complex dynamics of atmospheric methane, a potent climate change agent.

Reply: Thank you for the overall feedback on the present work. We thank you for the constructive comments. In the revised work, we have addressed all the suggested comments point-by-point carefully.

1.  Atmospheric methane ($CH_4$) is one of the high-potential greenhouse gases (GHG) that regulates the chemical reactions in the free troposphere and stratosphere. Comment: Here, 'regulates' does not seem appropriate.

    Reply: Thank you for the suggestion. We have modified the sentence as below and the same has been updated in the revised manuscript.

    Atmospheric methane ($CH_4$) is one of the high-potential greenhouse gases (GHG) and plays a vital role in the chemistry of the atmosphere. In the troposphere $CH_4$ oxidation is due to hydroxyl radical (OH) and results in the production of carbon monoxide, $CO_2$ and ozone in the presence of increased amounts of oxides of nitrogen where as in the stratosphere, oxidation of $CH_4$ is by OH radical, atomic oxygen and chlorine (Nair & Kavitha, 2020).

2.  Comment: Could figures 1a, 1b, and 1c be merged into a single figure where all the sources are indicated with different shapes/colors? Three heterogenic $CH_4$ source regions. Comment: Are the entities depicted in the figure referred to as heterogenic source regions or individual sources? Coal fields and thermal power plants are primarily situated within the same region.

    Reply: As suggested, Figure 1a, 1b and 1c are merged into single figure as shown below. The entities depicted in the figure are individual sources. The figure 1 below shows the top 10 wetlands (circle) which is natural sources of $CH_4$. The top 10 coal

mine locations (star symbol), top 10 thermal power plants (triangle) are anthropogenic sources of $CH_4$. The same has been updated in the revised manuscript.

[Figure]

3.  The Ministry of Environment, Forests and Climate Change (MoEF&CC), Government of India, has identified 75 Ramsar Wetland sites in India as of November 2022. These sites span a total area of 13,35,530 ha. Based on the high total geographical area coverage (Table 1), the top 10 places were determined for the current investigation. The size varies from 423000 ha (Sundarbans Wetland, West Bengal) to 18900 ha (Wular Lake, Jammu and Kashmir). Comment: Reference for the aforementioned statement.

    Reply: The reference for the aforementioned statements are given below:

    -   https://indianwetlands.in/wetlands-overview/indias-wetlands-of-international-importance/
    -   PIB Press Release on World Wetlands Day dated 26th August, 2022.

    The same has been updated in the revised manuscript.

4.  In the present study, the atmospheric $CH_4$ was obtained from 2009 to 2020…. Comment: As previously stated and indicated in Figure 2, data up to 2022 was utilized for the current study.

    Reply: Thank you for the suggestion. For studying the natural sources (wetlands) and anthropogenic sources (coal and thermal power plants) of $CH_4$, the $XCH_4$ concentration was taken for the period 2009 to 2022 is shown in figure 5. To maintain the uniformity in the datasets, we have updated the figure 5 with 2021 and 2022 years as shown below.

[Figure]

Figure 5. Monthly time series of $XCH_4$ over the a) wetlands, b) thermal power stations, and c) coal fields: sources of emissions, along with the overall growth rate at the respective site.

5.  Comment: There is no mention of gridding of level 2 data, if any has been applied. Have all the datasets been gridded to the same resolution?

    Reply: We have re-gridded the data to the same resolution

6.  Comment: In Figure 3, regarding the TROPOMI data from 2019 to 2022, is it averaged over this period or does it represent all the observations? There are data gaps in Tropomi, which are assumed to occur during the monsoon season. However, these gaps are absent in GOSAT. Are you employing any data-filling method for the GOSAT data?

    Reply: Figure 3c shows the averaged TROPOMI data from 2019 to 2022. As you pointed there are data gaps observed both in TROPOMI and GOSAT. The spatiotemporal variability of $XCH_4$ as discussed in figure 5 has few data gaps in the daily data during monsoon season which is due to the influence of cloud cover and sensor observation mode. However over the study locations a good number of representative $XCH_4$ data was observed in each month. Therefore we could analyse space time variability of $XCH_4$ over the study sites.

7. Figures 5a-c shows the monthly time series of $X$CH$_4$ over the specific sources of CH$_4$ dotted in the Indian region during 2009 to 2020. Comment: dotted?

   Reply: Thank you for the comment. Our intended meaning was different sources situated/pointed in the figure. However sentence has been modified now. The figures 5a-c shows the monthly time series of $X$CH$_4$ over the specific sources of CH$_4$ plotted for the Indian Region during 2009 to 2022. This figure is now updated with 2021 and 2022 data and same is reflected in the revised manuscript.

8. The seasonal cycle (peak and trough) of $X$CH$_4$ is strongly associated with the vegetation during the active phase of cultivation and reduced photochemical reaction by the hydroxyl radicals, respectively. Comment: could you provide a reference for the statement? Considering wetlands and coal as the largest emitter of CH$_4$, how is the seasonal cycle associated with this? A seasonal maximum of $X$CH$_4$ was observed over Coal and Thermal power plants from September to October and a minimum in pre-monsoon (March-May). Comment: Are you associating this cycle with rice cultivation?

   Reply: Sreenivas et al. (2016) in his studies observed high CH$_4$ concentrations during post-monsoon which may be associated with Kharif season. Low concentrations of CH$_4$ are observed during monsoon which is due to reduction in atmospheric hydrocarbons because of the reduced photochemical reactions and significant drop in solar intensity. As you rightly pointed seasonal maximum is also strongly associated with rice cultivation. Kavitha et al. (2016) also reported that different seasonal behaviour with seasonal peak in post-monsoon and low during monsoon in the Southern peninsular regions. These regional variations are to distribution of sources like livestock population, rice cultivation, wetland, biomass burning and oil and gas mining.

9. Comment: There is a typo in Figure 7, indi. Does the spatial distribution represent the mean of all the years?

   Reply: The typo has been corrected in figure 7 (indi to Indian Region).

10. Comment: The descriptions of Figures 7 and 8 do not match their respective figures. Additionally, the description of Figure 8 appears before that of Figure 7.

   Reply: Thank you. The figures numbers were wrongly written in the text of the manuscript. Figure 7 shows the continuous $X$CH$_4$ data from S5P/ TROPOMI complementing the GOSAT efforts instead of figure 8 which was written in the manuscript. In section 4.4, significant high emissions of CH$_4$ as shown in figure 9c, instead of figure 7c which was written in the manuscript. The figures numbers were corrected in the revised manuscript.

11. Significant high emissions of CH$_4$, as shown in Figure 7c. Comment: Figure 7c is missing.

Reply: The figure in section 4.3 is figure 9c instead of figure 7c and the same has been corrected in the revised manuscript.

**Additional Suggestions**

1. Emissions from EDGAR only verify the anthropogenic emissions, but there is a significant wetland source situated over India that requires verification with the appropriate inventory.

   Reply: We thank you for the important suggestion. We have implemented the emissions from the natural sources (wetlands) using the inventory "WetCHARTs v1.3.1". This has really helped us to apportion the contribution from the natural and anthropogenic emissions of $CH_4$. The below figure 8 shows the monthly time series of methane emissions (mg m$^{-2}$ month$^{-1}$) over the wetland sites, inset figure shows the seasonal methane emissions over the wetland sites from 2001 to 2019. As suggested, introduced new figure 8 in the revised manuscript.

2. Additionally, previous studies have demonstrated the limitations of satellite data during the monsoon season and the biases associated with global inventories such as EDGAR, over the Indian region. Therefore, it would be appropriate to study the uncertainty associated with the emissions and $X CH_4$ from their respective datasets.

   Reply: We agree with you suggestion. Present study included the relevant supporting information attributed to the uncertainties associated with the EDGAR based bottom-up $CH_4$ emission inventory and biases associated with the $X CH_4$ retrievals from the Remote Sensing sensors. However, this inventory helps to map the high emission (hotspots) zones of $CH_4$ and associated activities.

- Uncertainties in the information on source intensity, activity and other statistical data are the key parameters for the uncertainties in the EDGAR emission inventory (Janardanan et al., 2017).
- Bottom up inventory uncertainties range between 20 and 35% for agriculture, waste and fossil fuel sectors; 50% for biomass burning and natural wetland emissions and 100% or higher for natural sources such as geological seeps and inland waters for global methane emissions (Saunois et al.,. 2020).
- Ground based FTIR measurements of $X CH_4$ by the Total Carbon Column Observing Network (TCCON) are used extensively to validate the GOSAT retrievals. Retrieval bias and precision of column abundance from GOSAT SWIR observations have been estimated as approximately 15-20 ppb and 1% respectively (Morino et al., 2011; Yoshida et al., 2013).
- Methane retrieval from TROPOMI is in overall agreement with correlative ground based from TCCON and Network for Detection of Atmospheric Composition Change (NDACC). The systematic differences of the bias corrected $X CH_4$ data with respect to TCCON data and NDACC data are on average, -0.26±0.56% and 0.57±0.83% respectively (Song et al., 2023).

3. There are extensive studies of $X\text{CH}_4$ over India from satellites have been conducted by Mottungan et al. However, the authors have not reffered them in their study.

Reply: We sincerely appreciate the previous works carried out by Mottungan et al., which were missed in our work inadvertently. However, we could read those papers and utilised in the revised work to strengthen the present work.

---

## Author Response (AR2)

**General comments**

The study conflates emissions and concentrations by assuming a neat relationship between methane column concentration and emissions without consideration of other controlling factors: wind and the methane chemical sink. I recommend the authors discuss this question in a revised manuscript. OH trends are difficult to quantify, but observed concentrations can vary year to year based only on annual variations in the local wind speed, which is readily quantifiable. The authors could examine a reanalysis product (ERA5, GEOS-FP/MERRA-2) to assess trends in wind speed at their different source locations. This could support the argument that the observed trends in methane column mixing ratio are the result of emission trends.

Reply: Thank you for the suggestion and we totally agree with the points about the influence of winds and OH radical role in controlling the methane concentrations. As suggested, we have included and discussed about the wind trends in the revised manuscript computed at each source location. We have also discussed the chemical influence of OH radical, which acts as a strong sink for $CH_4$ concentration in the atmosphere.

- In the revised manuscript 10m u-component of wind from ERA5 which is reanalysis product is used for the period 2009-2022. The long-term seasonal plots (2009-2022) of wind speed ($ms^{-1}$) for coal and thermal power plants along with Long-term spatial trend of winds over the Indian region covering source types (coal and thermal power plants) for the period 2009-2022 are added in the revised manuscript.

The study should better address the very large uncertainties in the BU inventories it cites. The inventory estimates are not observations (despite several passages describing them as such, e.g., L. 29, 455), and EDGAR in particular has previously been shown to contain major spatial errors (e.g., see Maasakkers et al., 2023; https://pubs.acs.org/doi/10.1021/acs.est.3c05138).

Reply: We agree with your suggestion regarding uncertainties in bottom up inventories. In the present study we have used EDGAR inventory to understand the relationship between $XCH_4$ concentrations against anthropogenic EDGAR values. Maasakkers et al. (2023) reported annual gridded methane emissions inventory over the US at 0.1º×0.1º resolution while meeting the USEPA emission inventory standards which was submitted to UN in 2020. As suggested this article has reported the methane emission over the US region and has improved uncertainties compared standard global EDGAR database. Thanks for advising the article. Similarly, Solazzo et al. (2021) studied the uncertainties in Emissions Database for Global Atmospheric Research emission inventory of GHGs. The results indicate that the globally anthropogenic emissions covered by EDGAR for the combined three main GHGs for the year 2015 are accurate with an interval of -15% to 20%.

- $CO_2$ emissions which are responsible for 74% of the total GHG emissions account for ~ 11% of global uncertainty share.
- Wetlands are responsible for largest absolute uncertainty from all the $CH_4$ emission categories with a range of 107 Tg $CH_4$ $year^{-1}$, approximately 49.3% of the global total estimate. Large discrepancies exist in spatio-temporal variation of estimated $CH_4$

emission. One of the largest uncertainties in estimates of CH$_4$ emission comes from differences in wetland spatial extent (Zhang et al., 2017).

Specific comments

L. 29: No CH$_4$ flux is observed by WetCHARTS. Inventory values are reported, not observed.

Reply: It is corrected in the revised manuscript.

L. 57-60: What is the citation for these figures / the decadal budget?

Reply: Thank you for the suggestion. The citation for the above mentioned lines is given in the revised manuscript.

L. 146: The meaning of Ramsar is not immediately clear. I suggest writing "(see below)" or similar so readers know the term will be explained shortly.

Reply: Thank you for the suggestion. The same has been incorporated in the revised manuscript.

L. 147: How can a number of coal mines vary, and why is the range so large? Are these different estimates of coal mine count? If so, the large range is really surprising.

Reply: Thank you for the comment. The coal mines listed in the manuscript are based on the size and their production capacity. Details of coal mines and their classification are also provided in Pai et al. (2021) and Halder et al. (2024).

L. 185-188: These resolutions are not correct for the methane product. The methane product started at 7x7 km$^2$ resolution and was upgraded to 5.5x7 km$^2$ resolution at nadir (in contrast with the NO$_2$ product).

Reply: We agree with your suggestion. TROPOMI (TROPOspheric Monitoring Instrument) on board the Sentinel 5 Precursor satellite provides the methane product with its daily global coverage at a resolution of $7 \times 7$ km$^2$ since its launch in October 2017 and which is upgraded to $5.5 \times 7$ km$^2$ in August 2019. The resolutions are corrected in the revised manuscript.

L. 210: This be cited as Pai et al., not just a dataverse link.

Reply: Thank you for the suggestion. The same has been incorporated in the revised manuscript.

Figure 3 and related text: I assume the study region is shown as the blue box in Fig. 3b. Please clarify and include lat/lon ranges in text to facilitate reproduction of results and comparison with other work.

Reply: Thank you for the comment. The study region in the present work is Indian region covering the different sources (Coal mines, Thermal power plants and wetland sites). The figure 3c shows the probable high $X$CH$_4$ concentrations using TROPOMI data wherein 90$^{th}$

percentile statistical filter is applied. The blue box in figure 3c mainly covering Indian sub-continent (Latitude: 0°-40°; Longitude: 60°-100°).

L. 273-275: Please make these statements more quantitative. With respect to which year are the residuals defined? And how is the acceleration post-2015 diagnosed? The growth looks steady over the period to my eye.

Reply: Thank you for the suggestion. The lines 273-275 describes the anomaly/spatio temporal residuals in $XCH_4$ concentration which is calculated based on the individual data point for that particular year minus mean value for that particular year. Anomaly values in figure 3b range from -100 ppb to +100 ppb. Negative anomaly values in $XCH_4$ concentration indicate that particular data point is lower than mean value. Post 2015 the anomaly is on positive side. Based on your suggestion, a revised statement has been updated in the revised manuscript.

Figure 3c and related text: One would expect the tropics to show the highest columnar methane concentrations, because those columns are more sensitive to the troposphere than at higher latitude. I do not think these high concentrations can be said to reflect high emissions. Please clarify if I am misunderstanding what is being shown here.

Reply: Thank you for pointing out. The present study says that high concentrations of $CH_4$ are observed over the tropical region and do not necessarily indicate the high emission activities in the tropical belt. However, in studies by various researchers on methane emissions, Feng et al. (2023) used methane data from the Japanese Greenhouse gases Observing Satellite (GOSAT) estimate methane surface emissions, and results indicate relative to baseline value in 2019 largest annual increases in methane emissions during 2020 over Eastern Africa (14± 3 Tg), tropical South America (5± 4 Tg), tropical Asia (3±4Tg) and Temperate Eurasia (3±3 Tg). Further, in tropical and temperate South America, emissions increased by 9± 4 Tg and 4± 3 Tg, respectively.

Figure 4: This looks interesting, but the figure is qualitative and should be accompanied by concrete figures; otherwise, it's difficult to articulate the finding. Can you support the claim of "accelerated diffusion of $CH_4$ to SH" with numerical values? A clearer figure would plot the trend in $XCH_4$ by latitude (bin) over time.

Reply: Thank you for the suggestion. In the revised manuscript a new figure 4b is added which depicts the trend in $XCH_4$ globally using GOSAT data. $XCH_4$ has increased the global mean trend from 7 ppb year$^{-1}$ to 9 ppb year$^{-1}$. There are hotspots observed in the Tibetan Plateau (8.2 to 9 ppb year$^{-1}$), South America (8.2 to 8.8 ppb year$^{-1}$), African continent (8 to 8.4 ppb year$^{-1}$) and rest of the world $XCH_4$ trend varied from 6.7 to 8 ppb year$^{-1}$.

L. 311-313: Earlier in the manuscript, Parker is cited for the claim that seasonal $CH_4$ variation is dominated by wetland emission variability, but here the effect of seasonal OH trends is highlighted. Suggest discussing the roles of wetlands + OH in both passages.

Reply: Thanks for the comment and suggestions. The present paper discussed about the types of sources and probable sink mechanisms of $CH_4$. As suggested, we have discussed the

mechanism of OH radical availability in the troposphere and its role in removing the $CH_4$ from the atmosphere.

L. 374-379: These numbers all look very similar. What are the std's / error bars? Are they statistically different? At present I cannot tell if the results are strong enough for a meaningful trend comparison.]

Reply: Thanks for the suggestion. The below table shows the trend ± uncertainty for coal mines sites, thermal power plants and wetland sites calculated and same has been incorporated in the revised manuscript.

| Coal Mine locations | | | | | | | | | | |
|---|---|---|---|---|---|---|---|---|---|---|
| S. No | 1 | 2 | 3 | 4 | 5 | 6 | 7 | 8 | 9 | 10 |
| District names | Korba | Singrauli | Angul | Sonbhad ra | Jharsuguda | Dhanb ad | Paschim Bardha man | Bhadradri Kothagudem | Chatra | Cuddalore |
| Trend ±Uncertainty (ppb year $^{-1)}$ | 9.68 ± 0.52 | 9.4 ± 0.5 | 9.51 ± 0.5 | 9.4 ± 0.5 | 9.39 ± 0.52 | 9.78 ± 0.53 | 10.15 ± 0.55 | 9.5 ± 0.36 | 9.51 ± 0.56 | 9.13 ± 0.4 |
| Thermal power stations | | | | | | | | | | |
| Power station names | Vindhya chal STPS | Mundra TPS | Mundra UMPP | Sasan UMTPP | Tirora TPS | Rihans STPS | Sipat STPS | Chandrapur STPS | Anpara TPS | Korba STPS |
| Trend ±Uncertainty (ppb year $^{-1)}$ | 9.42 ± 0.50 | 9.69 ± 0.4 | 9.72 ± 0.41 | 9.46 ± 0.51 | 9.6 ± 0.51 | 9.42 ± 0.50 | 9.67 ± 0.50 | 9.66 ± 0.46 | 9.36 ± 0.50 | 9.67 ± 0.50 |
| Wetlands sites | | | | | | | | | | |
| Wetland locations | Sundar ban Wetland | Vembanad -Kol Wetland | Chilik a Lake | Kolleru Lake | Bhitarkani ka Mangroves | Point Calime re Wildlif e & Bird Sanctu ary | Loktak Lake | Upper Ganga River | Sambh ar Lake | Wular Lake |
| Trend ±Uncertainty (ppb year $^{-1)}$ | 9.54 ± 0.51 | 9.69 ± 0.44 | 9.5± 0.50 | 9.56 ± 0.45 | 9.32 ± 0.50 | 9.67 ± 0.45 | 9.58 ± 0.49 | 9.82 ± 0.52 | 9.52 ± 0.43 | 8.72 ± 0.3 |

Figure 8: I cannot read the text inset, and it is not clear to me what is learned from this figure. Is it saying that WetCHARTS doesn't show a trend in wetland emissions, whereas the satellites do? Again, it is essential to also examine potential trends in wind speed.

Reply: Thank you for the suggestion. The text inset is the 10 wetland sites and the figure depicts the seasonal methane emissions over the wetland sites. For better readability the inset figure is made into separate figure (8b) in the revised manuscript. A significant trend is observed over the Wular Lake with an increasing rate of 0.04 mg m$^{-2}$ year$^{-1}$ with a p value of 0.01. An annual trend of $X$CH$_4$ was over this study is about 8.72 ppb year$^{-1}$. As suggested, trends in wind speed are estimated using 10m u component from ERA5 reanalysis data for the period 2009-2022. The trend in wind speed over the wetland sites is given in the table 4 in the revised manuscript.

A positive trend over Wular lake (0.20 m/s) year$^{-1}$ and Point Calimere wildlife and Bird Sanctuary (0.25 m/s) year$^{-1}$ is observed.

L. 455: Again, no emissions are being "observed" here. The figure just shows how EDGAR values vary in time.

Reply: Thank you for the suggestion. This has been corrected in the revised manuscript.

Figure 10: What correlation is being plotted? Is this the correlation between annual TROPOMI vs. annual EDGAR per grid cell? Please clarify.

Reply: Thank you for the comment. The figure 12 shows the correlation between annual average TROPOMI vs annual average EDGAR per grid cell.

Corrections

L. 67: "contribute to 20-40%" -> "contribute 20-40%"

Reply: The same has been corrected in the revised manuscript.

L. 104: "Sentienl" -> "Sentinel"

Reply: The same has been corrected in the revised manuscript.

---

## Author Response (AR3)

**Response to reviewer**

Please consider addressing concerns from a reviewer, who made the following observations: "I had a chance to review the authors' changes. They've addressed my comments. My only remaining concern has to do with the results of their wind speed analysis. They found a decreasing wind trend for almost all of their hotspot/point-source sites where they report faster methane growth rate than average for the Indian subcontinent in the abstract. But they don't attribute that to emissions or wind (evidently the wind trend can explain at least part of it). So I'm just not sure what we're supposed to learn from this, but I don't have strong feelings about it. Perhaps other authors will find the analysis of methane concentrations and winds useful."

Reply: Thank you for the comment provided regarding wind speed analysis. A decreasing wind speed trend (m/s) year$^{-1}$ is observed over the majority of coal mine locations, thermal power plants and wetlands. In the present study, we attempted to address the $X$CH$_4$ concentrations, emissions and wind analysis. Similar attempts were also made by the Francis et al. (2023) and Chandra et al. (2017). Studies have reported CH$_4$ concentrations are column whereas emission processes are surface-based. An enhancement in the column CH$_4$ could be due to the advection by the background flow which acts as an important contributor. Also changes in boundary layer dynamics through convection process (vertical mixing) also play an important role in the column concentration.

As you kindly pointed out, we agree that negative wind trend is observed in the present study is expected to have slower winds which decreases the dispersion hence observed high CH$_4$ concentrations over the source locations. Also these trends are observed over the source location. Ricaud et al. (2014) studied the dispersion of the mid to upper level CH$_4$ levels due to the circulation with the Asian monsoon and found change in column CH$_4$ values.

References

Francis, D., Weston, M., Fonseca, R., Temimi, M. and Alsuwaidi, A., 2023. Trends and variability in methane concentrations over the Southeastern Arabian Peninsula. Frontiers in Environmental Science, 11, p.1177877.

Chandra, N., Hayashida, S., Saeki, T., & Patra, P. K. (2017). What controls the seasonal cycle of columnar methane observed by GOSAT over different regions in India?. Atmospheric Chemistry and Physics, 17(20), 12633-12643.

Ricaud, P., Sič, B., El Amraoui, L., Attié, J.L., Zbinden, R., Huszar, P., Szopa, S., Parmentier, J., Jaidan, N., Michou, M. and Abida, R.: Impact of the Asian monsoon anticyclone on the variability of mid-to-upper tropospheric methane above the Mediterranean Basin. Atmos. Chem. Phys, 14(20), 11427-11446, https://doi.org/10.5194/acp-14-11427-2014, 2014.